# An Experimental Study on the Drying-Out Ability of Highly Insulated Wall Structures with Built-In Moisture and Rain Leakage

**Klaus Viljanen [1,\*] and Xiaoshu Lu [1,2,3,4]**

1    Department of Civil Engineering, Aalto-University, FIN-02130 Espoo, Finland; xiaoshu.lu@univaasa.fi
2    Department of Electrical Engineering and Energy Technology, University of Vaasa, FIN-65200 Vaasa, Finland
3    Department College of Construction Engineering, Jilin University, Changchun 130026, China
4    Key Laboratory of Ministry of Land and Resources on Complicated Conditions Drilling Technology, No.968 Ximinzhu Rd, Chaoyang District, Changchun 130026, China
\*    Correspondence: klaus.viljanen@aalto.fi; Tel.: +35-85-0096-7873

**Abstract:** The recent research on highly insulated structures presents controversial conclusions on risks in moisture safety. This paper addresses these controversial issues through investigating the hygrothermal performance of energy efficient envelope structures under high moisture loads. The experiments consist of built-in moisture and rain leakage tests in mineral wool insulated structures. A heat and moisture transfer simulation model is developed to examine the drying-out ability in both warm and cold seasons. The results show that the energy efficient structures have an excellent drying out ability against built-in and leakage moisture. The difference in the drying ability is limited compared to conventional structures. A critical leakage moisture amount reaching the insulation cavity for a wood frame wall is determined to be between 6.9–20.7 g in a single rain event occurring every other day. Further research is required to target highly insulated structures, particularly addressing water vapor diffusion and convection.

**Keywords:** hygrothermal behaviour; built-in moisture; leakage; wood frame wall

## 1. Introduction

The energy efficiency of buildings has become increasingly important due to the greenhouse gas emission reduction and the acceleration of climate change. The endeavor for energy efficiency in the building sector has promoted the use of more effective thermal insulation in the building envelope to reduce conduction heat loss. The change in the building industry to use thicker insulation layers has raised the question of possible moisture problems in these structures. Mistrust in the hygrothermal behavior of highly insulated (HI) structures is not a new issue, since it was a topic already in the 1980s when there was a similar interest in buildings with low energy consumption [1].

The hygrothermal behaviour of energy efficient structures has been largely questioned [2–4]. In Gradeci's study of highly insulated walls with probabilistic methods, the hygrothermal performance was satisfactory for timber walls without potential deficiencies [2]. However, the performance decreased substantially with moisture sources as low as 0.5% of WDR (wind driven rain). Another study found that the drying ability might weaken due to the lower heat flow towards the outer part of structures [3]. The researchers added that a structure affected by an occasional moisture leakage is not able to dry out as fast as before. The study suggests that the overall fault tolerance is weakened with thicker insulation. The research was based on numerical simulations in selected reference years. A recent study of the hygrothermal behavior of highly insulated walls focused on the weather barrier material and its thickness, in the Norwegian climate [4]. The results

from this study imply that there is a risk for mold growth in highly insulated mineral wool structures with built-in moisture. The results further indicate that the increased thermal resistance of the weather barrier layer can reduce the risk for mold growth. These aforementioned studies imply that the risks involved with highly insulated structures are possibly connected to the high instantaneous moisture loads or significant levels of built-in moisture.

Other studies suggest that there are no significant risks with highly insulated structures [5–8]. Ojanen, for example, studied the influence of thermal insulation material on the drying efficiency of highly insulated wall structures [5]. His numerical study showed that a vapor open thermal insulation allows faster drying of built-in moisture compared to a more vapor tight insulation material. Moreover, Ojanen estimated that the drying potential (water vapor pressure difference) between the outer surface of the wall and the outdoor air does not change significantly between wall structures with thermal transmittances (U-values) 0.24 W/m$^2$K and 0.09 W/m$^2$K [6]. A study by Wang used a stochastic approach to simulate heat, air, and moisture (HAM) transfer in wall structures with high and conventional insulation levels [7]. The study used climate conditions from two cities in Canada. It showed that the mold growth risk is usually lower for a mineral wool wall with a 216 mm insulation than for a wall with 140 mm of mineral wool. To examine highly insulated structures, Pihelo simulated the hygrothermal behaviour of highly insulated timber frame walls in the Estonian climate [8]. He concluded that it is possible to design moisture safe timber frame walls with 400–600 mm insulation thickness. However, he emphasized that a careful selection of materials is required. These studies, that suggest there is no risk in highly insulated structures, are unsatisfactory because they lack experimental results on the drying ability of highly insulated structures under high moisture loads.

As the literature shows, there is no general agreement about whether the increase of the thermal resistance (R-value) in envelopes can lead to the degradation of their hygrothermal behaviour. Previous studies on highly insulated structures rarely deal with high instantaneous moisture loads. Moreover, there are few studies which concentrate on structures with only mineral wool as the insulation layer as many of the studies included other insulation materials [9,10]. Furthermore, many of the studies relied on computational analysis on HAM-transfer [2,3], which points to a need for more experimental studies to evaluate the performance of highly insulated structures.

In the evaluation of the hygrothermal behavior of highly insulated structures, it is essential to use correct, and high enough, moisture loads. The moisture sources usually considered as the most significant in the field of building physics are indoor moisture diffusion and convection, built-in moisture inside the structure, water leakages, and rainwater penetration into the structure. The magnitude of these phenomena varies greatly depending on the building type, structure, macro- and microclimate, airtightness of the structure, direction of the structure, and, finally, the moisture management during building stage. Therefore, a comprehensive evaluation of the influences of the envelope R-value on the structure's hygrothermal behavior needs to include all the significant moisture loads. These moisture loads must be compared to the drying-out ability of the structure and the hygrothermal conditions developed during the drying phase.

The present paper focuses on the effect of built-in moisture and rain water leakage on the hygrothermal behavior of energy efficient structures and compares it to a structure with less insulation. Furthermore, the study focuses on the drying-out ability of these structures. The study begins with material properties tests, followed by experimental laboratory tests. The two laboratory tests include prewetted structures and inserting moisture into the structures. The material property tests are used to evaluate both the moisture retention capability and drying-out capability, as well as to construct a hygrothermal simulation model. The simulation model is validated by comparing its results to the results acquired from the experimental tests.

## 2. Materials and Methods

### 2.1. Testing of Material Properties

The water vapor transfer properties were tested according to the European Standards EN ISO 12572:2016 and EN 12086 [11,12]. The materials used in the experimental studies in this and future articles were selected for the tests. Most of the samples had distilled water in the cup, as is recommended in [11] when using low vapor resistance specimens. However, the water vapor transfer properties at low relative humidity (RH) were tested by using a salt solution in a cup. Relative humidity of the ambient air was controlled by plastic environmental chambers with saturated salt solutions. The chambers were in a weather room with an average temperature of 24 °C and an average relative humidity of 45%. The size of each chamber was $40 \times 70 \times 15$ cm$^3$. Air was constantly circulated in each chamber with a fan. The correction for the effect of a masked edge of the specimen was made for the mineral wool and gypsum board samples according to EN ISO 12572:2016. The correction for resistance of air layers was made according to EN ISO 12572:2016.

The hygroscopic sorption properties were tested according to the European Standard EN ISO 12571 [13]. The material samples were kept in similar humidity chambers as in the water vapor transfer property tests. Saturated salt solutions and air circulation were used to ensure an even relative humidity inside the chambers. The sorption was first tested for the adsorption phase, followed by the desorption phase. The test begun and ended by drying the samples in an oven at a temperature of 105 °C. The sample sizes were the following: Wood and gypsum boards of 0.1 dm$^3$, mineral wools of 1.1 dm$^3$, and wind barrier mineral wools of 0.5 dm$^3$.

### 2.2. Test Set-Up for Built-In and Leakage Moisture in Wall Structures

The built-in moisture test aimed to explore the drying rates of initially humid wood frames in wall structures. The wood frames from Nordic softwood were first moistened in a weather room with active water sprinkling and a relative humidity close to 100%. The vertical moisture flow in the humidity room was measured with the mass change of a paper towel. The moisture flow was approximately 0.3 mm/h. The humidification period lasted 16 days, after which the wood frames had an initial average moisture content of 26–28 percent by weight. The humidification period is equivalent to a rainy and foggy building phase. The drying phase of the built-in moisture test was assembled into an Arctest-1500 environmental test chamber.

The test included four wood frame wall structures for which wall 1 served as a baseline wall (BL) and the others as highly insulated walls (Table 1, Figure 1). The selected HI wall compositions are common in Finland. The baseline wall was a common structure in Finland until the year 2010. The wood frames were initially cut precisely to fit exactly into the wall framework, leaving a minimal gap against the plywood frame. The plywood frame was coated with a polyethylene (PE) foil on all sides to prevent moisture transfer into the frame. Thereby, the test walls corresponded to high wall sections with little vertical moisture transfer. The indoor conditions on the warm side of the experimental walls was constructed with a PE-foil assembled in a wood frame. The ventilated cavity and the facade cladding material was omitted from the test structures. Thereby, the test represented structures with adequate ventilation behind the facade cladding.

**Table 1.** Hygrothermal properties of the test walls in the drying phase of the built-in moisture test. The Sd-value stands for the water vapor diffusion-equivalent air layer thickness.

| Test Wall | U-Value [W/m$^2$K] | Sd-Value of Materials Outside Wood Frame [m] | Sd-Value of Vapor Barrier [m] | Added Moisture During Humidification [g] |
|---|---|---|---|---|
| BL (baseline) | 0.22 | 0.073 | 76.5 | 1096 |
| HI 1 | 0.12 | 0.099 | 76.5 | 809 |
| HI 2 | 0.12 | 0.099 | 0.3–25 | 838 |
| HI 3 | 0.12 | 0.172 | 76.5 | 806 |

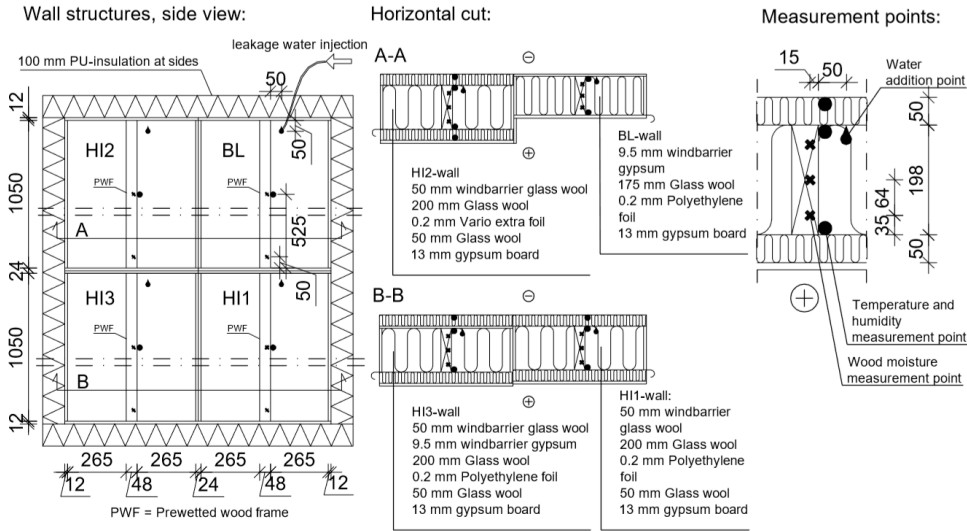

**Figure 1.** The test setup in the built-in moisture and leakage moisture tests.

Both indoor and outdoor environments were equipped with thermo- and hygrostats to create a specific temperature and humidity condition. The moisture content of the wood frame was measured from a total of 24 points with Scanntronik's (Zorneding, Germany) Material Moisture Gigamodule and Thermofox equipment, measuring moisture content (MC) based on the material's electrical resistance. The measurement depth was selected to be 10–20 mm from the wood surface. The measurement method results in the maximum moisture content from the measurement area. The precision of the equipment is 1 weight-% below 85% relative humidity and between 85–100% relative humidity the precision is 1–4 weight-%, as the precision declines somewhat linearly towards 100% relative humidity. The equipment measurement range is 6–90 weight-%. The MC as weight-% was calculated from the electrical resistance according to [14], considering the wood species and temperature.

Temperature and humidity were measured, from the middle height and 5 cm upwards from the bottom of the test walls, with Vaisala HMP44 probes. The accuracy of the relative humidity measurement is ±2% between 0–90% relative humidity and ±3% between 90–100% relative humidity. The probes were calibrated in salt solutions before the measurements.

The leakage moisture test was performed to evaluate the drying ability of the wood frame wall structures exposed to the infiltration of rainwater into the stud space. This test was performed with the same test structures as in the built-in moisture test (Figure 1). The leakage test was started after the built-in moisture had dried off. It was important to achieve an equivalent moisture load in all four test walls to be able to equally evaluate the drying out capability of the structures. Therefore, the water insertion system, shown in Figure 1, was designed in a way which made it possible to add the same amount of water in each test wall during every irrigation event. The water insertion was positioned in each wall structure, correspondingly, to represent rainwater leakage into the thermal insulation layer behind the exterior material layer(s). The pointwise insertion position is depicted in Figure 1, in the side view and in the horizontal cut. The insertion method has the benefit that the added moisture will retain in the structure and not run off, as would occur if the water was added, e.g., in either of the surfaces of the ventilation cavity. The water was injected into the structures with a PVC-hose and hand pump, which were calibrated for the presented accuracy. The air pressure in the hose removed the remaining water droplets from the inner surface of the hose. The insertion method is similar to the one used in [15].

The amount of leakage moisture was evaluated based on the rain statistics in Helsinki during years 2010–2018 [16] and rain measurements in Finland during 2000–2002 [17]. The month with most rainfall in Helsinki (93 mm) is August, whereas the average rainfall per month is 58 mm [16]. The total rainfall per year is, on average, 712 mm [16]. The dimensioning rainfall per month was selected to be

150 mm, which has only been exceeded in two months during the second decade of the 21st century. During the summer months, the average rain time is 6.17% (44.4 h) [17] and the average long-term rain duration is four hours [16]. The horizontal rainfall is thereby 3.4 mm/m$^2$h. The vertical rainfall on the building facade was calculated from the horizontal rainfall with the equation presented by Straube and Burnett [18] to be 1.15 mm/h. Here the rain admittance factor (RAF) was set to 0.5, the driving rain factor (DRF) set to 0.225, wind speed set to 3 m/s, and the direction of wind set to 0 deg. These values correspond to average conditions during rain in the middle of the facade. The ratio of water leakage to water spray rate in a previous laboratory experiment was between 0.1 and 6% [15]. The amount of water directly injected into the thermal insulation space was selected to be 1% (0.0115 kg/m$^2$h) of the vertical rainfall impinging the building facade. Considering the test wall size of 0.6 m$^2$, the injected water amount was 0.0069 ± 0.0001 kg per wall every 38 h, on average (phase one). The rain leakage test was started with a small amount of water to avoid water outflow from the test structures, which has been the case in a previous study [19]. In the case of water outflow, the drying-out ability of each structure could not be evaluated precisely. This risk of leakage water outflow from the structures was somewhat increased by the relatively small size of the test specimen (wall section, 0.6 × 1.0 m$^2$). One might argue that, in wood frame structures, the height of the test structure is not imperative to the drying-out ability since most of the water would accumulate in the lower part of the test structure. Thereby, the drying-out ability would be determined by the properties and dimensions in the lower part of the structure. The injected water amount was subsequently triplicated to 0.0207 ± 0.0001 kg per wall every 67 h, on average (phase two) based on the measurement results, and this amount was considered significantly larger than in stage one, yet moderate. The total leakage amount was 0.1035 kg in phase one and 0.3135 kg in phase two.

### 2.3. Test Set-Up for "Drying Cakes"

The drying ability of highly insulated wall structures was further evaluated with a test including cylinder shaped test pieces with a high level of built-in moisture (Figure 2). Wood disks from Nordic softwood with a diameter of 100 mm and height 23–24 mm (half the thickness of a typical frame) were used. The same wood species were used as in Section 2.2 in order to represent the wood frame as the source of the built-in moisture. The disks were sealed vapor tight from all surfaces, except the other circular surface with liquid waterproofing (Kerafiber). After this, the wood disks were moistened in the same weather room as was used in Section 2.2. After the humidification period, circular gypsum board and wind barrier mineral wool pieces were added on top of the wood disks to form the "drying cakes". The cakes were further sealed vapor tight with tape. The Sd-value of the sealing tape was determined, with a cup-test, to be approximately 25 m. Thus, the drying of the cakes was primarily possible towards the outdoor air. The wind barrier material (gypsum board or mineral wool) had an open surface with a diameter of 73 mm.

Each cake was removable from the larger test wall via the outdoor air climate chamber. The test wall had variable thickness to account for the different U-values of the structures the cakes represented. The drying ability of the structures was estimated based on the mass change of the cakes. The cakes were weighed every four days, on average, after which they were reassembled into the test wall. Heat flux was measured from behind each cake by attaching heat flow sensors to the sample with a heat conductive tape. For this, Hioki heat flow sensors (Z2015-01) were used, which have a repeatable precision of 2%. The heat flow sensors were equipped with thermocouples to measure also the temperature. The outdoor air temperature was set to 5 °C, −10 °C and −2 °C during the test in phases 1–3. The indoor air temperature was 19.9 °C and the outdoor relative humidity was 77.5%, on average.

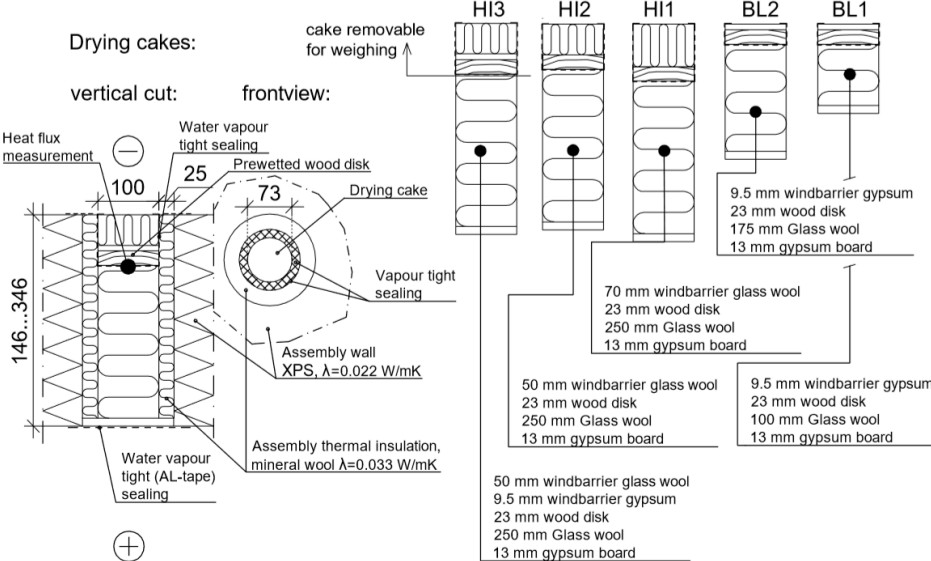

**Figure 2.** The test setup in the drying cakes test.

### 2.4. Simulation Model

A numerical simulation model of different wood frame wall structures was developed. The aim of the model was to simulate the drying process of structures with different insulation levels. The model represented a horizontal cut of the wall structures (Figure A1). Thereby, the model was two-dimensional, which was evaluated as an appropriate dimension for the drying of a wood frame with no vertical change in the moisture content. The model was implemented into a commercial simulation software, Comsol Multiphysics 5.3a. The moisture transfer was modelled by water vapor diffusion, which is the most common method to simulate problems related to moisture transfer in wood [20].

The material properties from the Section 2.1 tests were used and the remaining material properties were selected from the literature and the documents provided by the material manufacturers. The hygrothermal properties of the materials are presented in Table 2. The initial moisture profile in the wood frames could not be evaluated based on the weight change of the frames during humidification because the major part of the weighted additional moisture was located at the ends of the frame. Instead, the initial moisture profile in the wood frame was evaluated based on the wood moisture measurements performed for an additional wood frame after the weather room humidification described in Section 2.2. As a result of these MC-measurements, the wood frame was modeled with three domains. The domain thicknesses from the frame core was 10 mm, 4 mm, and 10 mm with corresponding initial MC-values of 18.7 weight-%, 22.6 weight-%, and 30.0 weight-%. In HI5–HI6 walls (Figure A1 and Table A1), the initial condition of the wood frame was different at the cold and warm side of the frame. The conditions for the moist frame were set to the outmost 200 mm part of the frame. An additional moisture barrier with an Sd-value of 2 m was set between the moist part and the inner part of the frame. Therefore, the initial moisture content of the deep cavity walls would present an equilibrium in capillary moisture levels. In practice, this kind of moisture content in the outer part of the frame could result from the building stage with poor weather protection. The warm side of the wood frame in HI5 and HI6 walls and the other materials in the wall structures had an initial relative humidity of 60%.

The simulation model was compared to the experimental results from the built-in moisture test. After this, the drying of the built-in moisture was simulated for the structures BL and HI1–3 from the drying test and some additional structures that were not included in the experimental tests (Appendix A). The drying phase was simulated for realistic weather conditions in Finland. The moisture reference year, "Vantaa 2007" [21], was used as the outdoor climate conditions, which

represent the current Finnish climate. Solar radiation, moisture from rainwater, and thermal radiation were omitted from the simulation, considering structures with high air exchange rates in the ventilation cavity. The climate data was averaged over 24 h periods to speed up the numerical solution process.

**Table 2.** Heat and moisture transfer related material properties used in the simulation. GW stands for glass wool and wb for wind barrier.

| Material | $\lambda$ [W/mK] | $c_p$ [J/kgK] | $\mu$ [-] |
|---|---|---|---|
| GW | 0.033 | 850 | from Section 3.1 |
| GW (wb) | 0.031 | 850 | from Section 3.1 |
| Gypsum | 0.21 | 1100 | from Section 3.1 |
| Gypsum (wb) | 0.21 | 870 | from Section 3.1 |
| Plywood | 0.11–0.13 | 1500 | 25–232 |
| Fiber cement board (FCB) | 0.34 | 840 | 17–43 |
| Nordic softwood | 0.129–0.191 [1] | 1880 | from Section 3.1 |

[1] Depending on the moisture content as presented in [22].

The outdoor weather, mainly temperature and relative humidity, can affect the drying rates of the structures. Therefore, the beginning of the simulation was scheduled for both warm (first simulation period "summer-winter", with duration 10.6–25.1) and cold seasons (second simulation period "autumn-spring", with duration 9.10–26.5). The indoor moisture excess was 5 g/m$^3$ below temperatures of 5 °C and 3 g/m$^3$ above temperatures of 15 °C, as presented in Finnish guidelines [23] for residential and office buildings. Linear interpolation was used between these temperatures.

The mould growth risk in the structures during the drying phase was evaluated using the Finnish mould growth model [24,25], a model based on the original work presented in [26]. Hourly values of temperature and relative humidity on the cold side of the wood frame surface, and 20 mm from the frame on the rigid board surface, were used as an input for the model. The most common wood type for a wooden frame in Finland is planed timber. Thereby, the mould index (MI) was calculated with the following parameters: Growth rate sensitivity 2 (sensitive), sensitivity for maximum amount of mould 2 (sensitive), and recession class 0.25 (moderate recession). The same values applied for the rigid board from plywood and gypsum. For the fiber cement board, the values were the following: Growth rate sensitivity 3 (moderate sensitive), sensitivity for maximum amount of mould 2 (moderate sensitive), and recession class 0.1 (low recession).

## 3. Results

### 3.1. Material Properties

The results from the hygroscopic moisture test are presented in Table 3. The results from the water vapor transfer properties tests are presented in Table 4. The calculated sorption curves contained some inconsistent results for mineral wool products. In particular stone wool (SW) products showed marginally lower moisture contents for desorption than for adsorption. A probable reason for this was the low hysteresis effect of stone wool combined with the mass loss of the specimen during the test. The dry weight before and after the test revealed that the mass loss for stone wool products was 0.09 kg/m$^3$ and 0.04 kg/m$^3$ for glass wool products. These levels of mass loss affect the consistency of the results for stone wool but not for glass wool. The hysteresis for glass wool was larger compared to stone wool. The water vapor diffusion resistance factor of Nordic softwood was strongly dependent on the relative humidity.

**Table 3.** Calculated water vapor diffusion resistance factors (μ) and Sd-values of the coatings based on the measurement results from the cup-tests [12].

| Material | Density [kg/m³] | Relative Humidity [%] | | | | |
|---|---|---|---|---|---|---|
| | | 24.5 | 39 | 57 | 72 | 93 |
| | | μ [-] and Sd [m] | | | | |
| GW | 23.7 | 1.1 | 1.3 | 1.7 | 1.6 | 2.1 |
| GW (wb) | 68.6 | 1.2 | 1.1 | 1.5 | 1.4 | 1.9 |
| SW | 30.9 | 1.3 | 1.3 | 1.3 | 1.2 | 2.1 |
| SW (wb) | 96.4 | 1.1 | 1.3 | 1.3 | 1.2 | 2.1 |
| GW coating (Sd) | - | 0.03 | 0.02 | 0.01 | 0.01 | 0.02 |
| SW coating (Sd) | - | 0.06 | 0.04 | 0.02 | 0.04 | 0.04 |
| Gypsum (wb) | 759.2 | 5.2 | 5.4 | 7.4 | 7.4 | 8,0 |
| Nordic softwood | 513.5 | 289.7 | 94.5 | 29.2 | 25.1 | 7.9 |

**Table 4.** Hygroscopic moisture [kg/m³] as a function of relative humidity measured according to [13]. Ads stands for adsorption phase and des for desorption phase.

| | | Relative Humidity [%] | | | | | | | |
|---|---|---|---|---|---|---|---|---|---|
| Material | Phase | 11 | 33 | 45 | 62 | 75 | 85 | 89 | 97 |
| GW | ads | 0.29 | 0.33 | 0.29 | 0.39 | 0.39 | 0.63 | 0.64 | 2.19 |
| | des | 0.35 | 0.40 | 0.36 | 0.46 | 0.46 | 0.69 | 0.72 | |
| GW (wb) | ads | 0.34 | 0.45 | 0.47 | 0.64 | 0.72 | 0.86 | 1.25 | 4.48 |
| | des | 0.48 | 0.62 | 0.68 | 0.83 | 1.01 | 1.33 | 1.46 | |
| SW | ads | 0.21 | 0.21 | 0.23 | 0.25 | 0.24 | 0.25 | 0.25 | 0.27 |
| | des | 0.19 | 0.21 | 0.23 | 0.23 | 0.23 | 0.26 | 0.24 | |
| SW (wb) | ads | 0.32 | 0.35 | 0.41 | 0.42 | 0.45 | 0.45 | 0.47 | 0.5 |
| | des | 0.31 | 0.37 | 0.40 | 0.39 | 0.44 | - | 0.47 | |
| Gypsum | ads | 1.1 | 2.4 | 3.2 | 4.1 | 5.6 | 7.3 | 9.8 | 18.5 |
| | des | 4.3 | 5.7 | 6.6 | 8.0 | 9.0 | 10.6 | 11.2 | |
| Gypsum (wb) | ads | 1.3 | 2.5 | 3.3 | 4.3 | 5.6 | 7.2 | 10.1 | 18.7 |
| | des | 4.2 | 5.6 | 6.7 | 7.7 | 9.1 | 10.7 | 11.7 | |
| Nordic softwood | ads | 19.2 | 33.7 | 40.9 | 56.3 | 69.1 | 91.5 | 98.5 | 130.4 |
| | des | 21.4 | 39.7 | 50.0 | 66.3 | 83.4 | 108.1 | 116.0 | |

*3.2. Results from the Built-In Moisture Test*

The moisture distribution of the wood frames after the humidification phase was evaluated with an additional wood frame. The wood moisture of this piece was measured after the humidification with a handheld wood moisture meter (Gann hydromette HT 75). The moisture content of the frame was 19–20 weight-% in the frame core, 21–24 weight-% 1 cm from the outer surface, and 25–34% 0.2 cm from the surface. Consequently, the moisture distribution of the frames was ascending towards the outer surface.

The climate conditions during the built-in moisture test are presented in Figure 3 with the corresponding average values for the test period. In addition, Figure 3 presents measurement results from temperature and relative humidity in the structures. The measured RH varied significantly depending on the structure and measurement point. The highly insulated walls HI1–HI3 had largely equal rates of decrease in the RH values in the cold side of the wood frame. Here, the relative humidity reached a level of 60–65% in 42 days, with the HI3 wall reaching the 65% value. This result indicates that the gypsum board behind the wb-mineral wool in wall HI3 has a small retarding effect on the drying out ability of the wall. For the baseline wall the RH settled to a level of 80–85% during the first 42 days, which is probably connected to the lower thermal resistance of the gypsum board compared to the wind barrier mineral wool used in HI1–3 walls. The changes in the RH-values during the rest of the test (62 days) were minimal, indicating that most of the built-in moisture had dried from the structures. Measurements from the warm side of the wood frame showed great resemblance in the RH-curve form.

However, the RH-level in the baseline wall was 4–20% lower during the test. This notable difference in the RH level, especially in the drying phase, is probably caused by the lower thermal resistance inside the wood frame in the BL wall compared to the HI walls. Most of the RH level change took place during the first 56 days of the test. Measurements behind the wind barrier coating in walls HI1–HI3 showed relatively similar values during the drying phase of the wood frames. The RH was slightly above outdoor level for walls HI1 and HI2 in the beginning of the test, which denotes an elevated moisture flux towards the outdoor air. This is caused by the lower Sd-value of the wb-layer in walls HI1–2 compared to wall HI3. After the drying phase, the RH in the outdoor air was higher than in the test walls. This suggests that the moisture flux was in such a low level that it did not increase the RH behind the wind barrier coating or the wind barrier gypsum. In addition, wall HI2 showed 2–3% higher RH-values than walls HI1 and HI3 after the drying phase, which implies that the moisture flux through the vapor barrier in wall HI2 was higher than in the other walls. The Sd-value for the PA-foil used in the HI2 wall is approximately 25 m, with 35%-RH. Thereby, the difference in the RH values behind the wind barrier coating, resulting from indoor air water vapor diffusion, was limited.

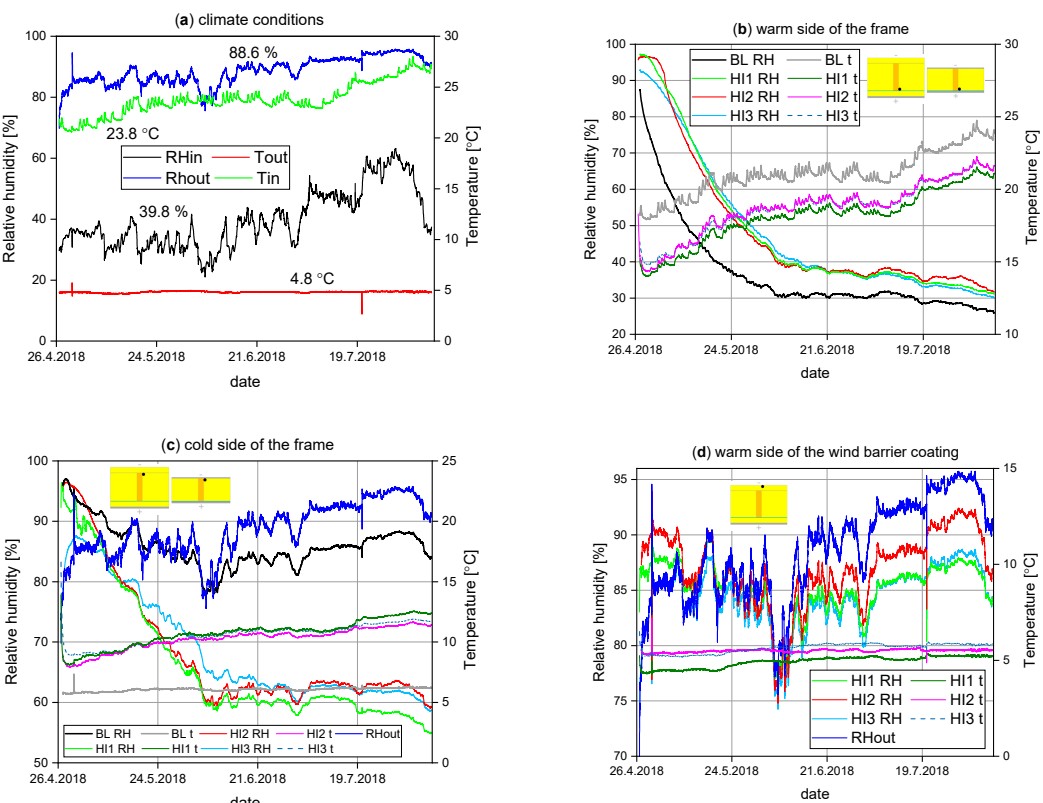

**Figure 3.** (**a**) the climate conditions; (**b**–**d**) measured relative humidity and temperature during the built-in moisture test. Measurement locations are described in the subfigure headings.

The moisture content of the wood frame during the built-in moisture test is presented in Figure 4. For all the walls, the moisture content at the cold side of the wood frames at the middle height of the frame declined from the level of 30–33 weight-% to a level of 13–15 weight-% during the first 50–55 days. The drying rates of walls HI1 and HI2 were similar, as well as between walls HI3 and BL. The drying rate for walls HI1 and HI2 was higher than with the HI3 and BL walls, although the difference was minor. The same difference was observed from the RH values behind the wind barrier coating in the beginning of the drying phase. RH values were higher in walls HI1 and HI2 compared to wall HI3.

At the warm side in the middle frame, the moisture contents reached the same level as the cold side during the first 26 days. The highest drying rate was observed for the BL wall and the second

highest for wall HI2. In the BL wall the high drying rate might be connected to the higher temperatures in the warm side of the wood frame and in the HI2 wall this might result from the more vapor-open PA-vapor barrier.

The MC results from the bottom of the walls were relatively similar compared to the middle height of the frame. However, the BL wall showed a slower drying rate in the cold side of the wall. This is probably connected to the sealing of the test walls and the highest initial moisture content (over 35 w-%) for the BL wall at that point of the wood frame. The effect of the sealing tape is discussed further in Section 4. Other MC results from the cold and warm side of the bottom frame showed a high resemblance as was observed for the corresponding RH curves. The drying rate in the warm side of the frame for the BL wall was not higher compared to the HI walls, as was observed in the middle height. The drying ability in such details of a wall is lower than in the middle height of the wall. Therefore, small differences in the drying ability might equalize.

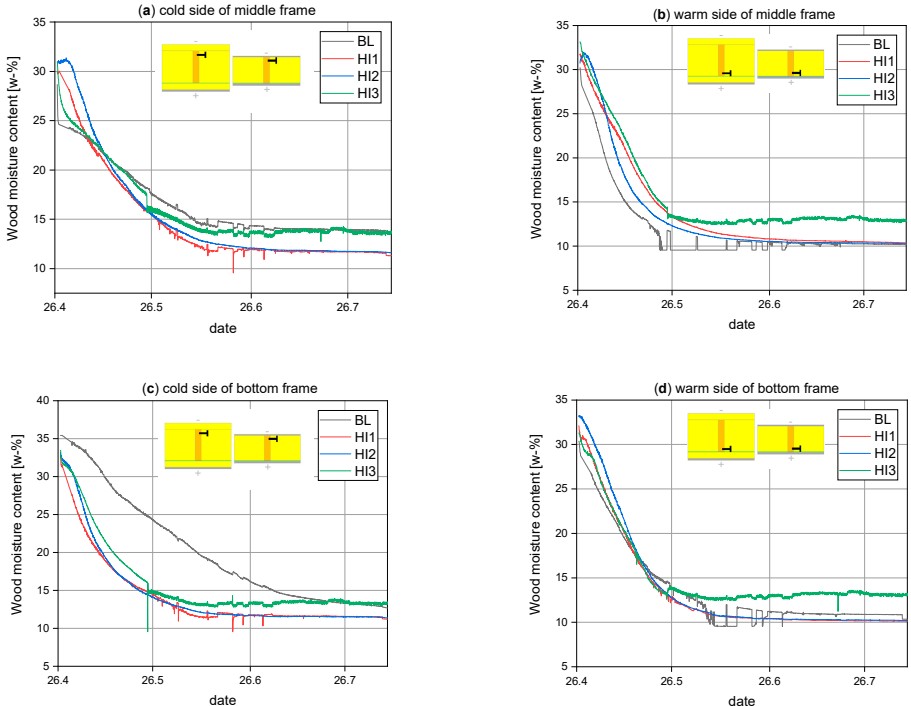

**Figure 4.** Measured moisture content of the wood frame (**a**–**d**) during the built-in moisture test. The measurement locations are described in the subfigure headings.

### 3.3. Results from the Drying Cakes Test

The wetting phase of the wood disks used in the drying cakes test took 22 days (Figure 5a). The weight change of the wood disks was relatively similar between the samples. On average, 51% of the total water uptake occurred during the first 24 h. In total, the water uptake was 30–44 g for the wood samples, with an average weight of 76 g. However, the moisture transfer rate into wood disk HI2b was slower than for the other 9 samples. Only 29% of the total water uptake occurred during the first 24 h, with the total uptake of 30 g. The final moisture uptake for disk BL2b (44 g) was higher than for the other samples, although only 41% of the total water uptake took place during the first 24 h. The conditions in the humidification room were similar for all the wood disks. Therefore, differences in the sample weight change are probably connected to the different moisture intake properties of the wood disks. The density of the wood disks did not differ significantly (388–422 kg/m$^3$).

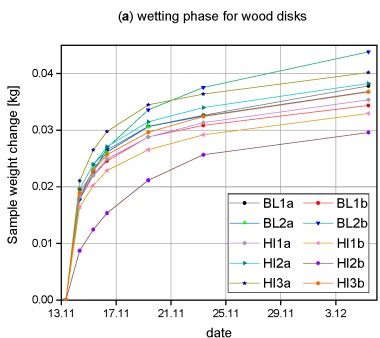 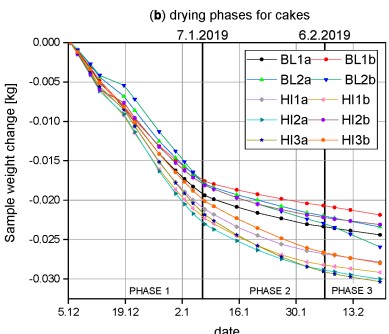

**Figure 5.** (**a**) wetting phase for the wood disks; (**b**) drying phase for the cakes in the drying cakes test.

The climate conditions can have a significant effect on the drying-out ability of the test samples. The outdoor relative humidity during the drying cakes test was, on average, 77.5%. Thereby, the average relative humidity in the materials outside the wood disks was 89%, assuming the relative humidity on the outer surface of the wood disk to be near 100%. Based on the results from Section 3.1, the Sd-value for the baseline samples in the early stages of the test was 0.08 m and 0.12–0.2 m for highly insulated samples.

The heat flux measurements showed that the test assembly influenced the theoretical heat flow rates during the drying cakes test. The U-values, that were calculated from the heat fluxes and measured climate temperatures, were larger than the theoretical U-values (Table 5). The reason for the increased heat flux through the cakes was the increased temperatures at the warm side of the cakes. This was confirmed with the thermocouples in the heat flux sensors, which showed 2 °C higher temperatures for the BL-cakes and 5 °C higher temperatures for the HI-cakes, compared to the steady state temperature calculated with the thermal conductivities of the materials. The heat flux through the HI walls was lower compared to the BL walls. The average U-value for the HI walls was 0.18 W/m$^2$K and for the BL walls was 0.36 W/m$^2$K.

**Table 5.** The results from the heat flux measurements and hygrothermal properties of the test samples. The calculated U-value based on the measured heat flux is in the parenthesis. $R_{wb}/R_t$ stands for the percentage of thermal resistance, which is located outside the wood frame.

| Sample Type | U-Value [W/m$^2$K] | Sd-Value 71% [m] Outside Wood Disk | Sd-Value 93% [m] Outside Wood Disk | $R_{wb}/R_t$ [%] |
|---|---|---|---|---|
| BL1a | 0.29 (0.44) | 0.070 | 0.076 | 1.2 |
| BL1b | 0.29 (0.43) | 0.070 | 0.076 | 1.2 |
| BL2a | 0.17 (0.31) | 0.070 | 0.076 | 0.7 |
| BL2b | 0.17 (0.26) | 0.070 | 0.076 | 0.7 |
| HI 3a | 0.104 (0.21) | 0.149 | 0.195 | 17.2 |
| HI 3b | 0.104 (0.20) | 0.149 | 0.195 | 17.2 |
| HI 1a | 0.098 (0.16) | 0.107 | 0.158 | 22.0 |
| HI 1b | 0.098 (0.12) | 0.107 | 0.158 | 22.0 |
| HI 2a | 0.104 (0.19) | 0.079 | 0.119 | 16.8 |
| HI 2b | 0.104 (0.20) | 0.079 | 0.119 | 16.8 |

The amount of dried moisture during the drying cakes test was evaluated for each wall (Figure 5b). In the beginning of phase one, with an outdoor temperature of 5 °C, the HI walls showed 20–35% faster drying compared to the BL walls. The difference in the drying speed declined towards the end of phase 1, reaching a level of 16% faster drying. The BL1 walls dried faster than BL2 walls, especially in the beginning of phase one. The drying rate in wall BL2b was lowest of all the BL walls. Wall BL2b had the lowest measured U-value among the BL walls. Similarly, BL1a had the highest measured U-value and the highest drying rate among the BL walls. These results suggest there is a small negative correlation in the drying rate and the insulation level in structures with a low R-value in the wind barrier layer.

The drying rates among walls HI1–3 were not higher for any wall type, implying that the additional gypsum board behind the wb-insulation or the increase in the thickness of the wb-insulation from 50 mm to 70 mm does not affect the drying rate of the HI wall structure.

For phase two, the outdoor temperature was lowered from 5 °C to −10 °C. In phase two the drying rate was 15–21% faster for walls HI1–3 compared to walls BL1–2. The cold outdoor temperature reduced the drying rates more for the BL walls than for the HI walls. The negative correlation between drying rate and insulation level found in walls BL1–2 in phase one was not detectable at phase two.

In phase three the temperature was kept at −2 °C. The drying rate remained 17–21% faster for the HI walls. Wall BL2b showed fast drying in the end of the test. This might result from the highest additional moisture level in this wall after the humidification. Therefore, this sample could have had additional moisture while the other samples had somewhat reached an equilibrium in moisture content in the structures. The final weight change of sample HI2b was with the same level as the BL samples while other HI samples had a 21.6% higher final weight change, compared to the BL samples.

Throughout the test the drying rate among walls HI1–3 was not dependent on the varied structural properties (wb-layer thickness and gypsum board behind wb-mineral wool). The weight change results from the drying cakes test suggest that the higher thermal resistance of the wb-layer increases the drying ability of the structure.

### 3.4. Results from the Rain Infiltration Test

The results of the relative humidity measurements in the mineral wool during the rain infiltration test are presented in Figure 6. The conditions during test are presented in Figure 7. Average values in the climate diagram are indicated with numbers. The result figures include vertical lines to indicate the duration of the test phases. Phase one, with a recurrent leakage moisture load of 6.9 g, lasted 22 days. Phase two, with a recurrent leakage moisture load of 20.7 g, followed phase one with a duration of 40 days. The final, phase three, with no leakage moisture lasted 47 days. The leakage events are indicated as dots in the lower part of the figures.

The form of the relative humidity curves was similar for all wall types in the middle height of the frame, on the cold side of the wall (Figure 6a). However, the changes during the water leakage event were significantly larger for wall HI1; the relative humidity jumped from 60–80%, whereas for the other wall types this change was typically below 5%. Wall HI2 has a more vapor-open PA-foil vapor barrier, whereas HI1 has a PE-foil vapor barrier. Walls HI3 and BL have hygroscopic gypsum board against the cavity insulation, whereas wall HI1 has wb-mineral wool with low hygroscopicity at that point. These differences might affect the changes in RH in the short term. For the baseline wall the relative humidity varied between 70–80% and for the HI walls between 55–70%, disregarding the peaks for the HI1 wall. The R-value of the structure outside the wood frame is remarkably lower for the BL wall than for the HI walls, which affects the RH levels through occurring temperatures. During phase two, the average relative humidity in the HI1–HI3 walls rose from 60–65%. This implies that the leakage moisture amount of 20.7 g was high enough to surpass the moisture retention capabilities of the materials affected by leakage. For the BL wall the change in the initially higher RH level was minimal. With higher initial relative humidity, the BL wall has a higher slope in the hygroscopic moisture curve of the gypsum board. Thereby, excess water from leakages is absorbed by the gypsum. In phase three it took roughly two weeks for the HI walls to reach the same RH levels as before the leakage test. This implies that the structures had a high drying ability against moisture loads from rain infiltration. The RH level in wall HI1 was 3–5% lower than in other HI walls, which was also observed in the built-in moisture test. The reason for the RH difference might be connected to the hygroscopic gypsum board in wall HI3 and the PA-foil in wall HI2. A closer look at the RH curves in the middle height at the cold side of the frames during phases one and two (Figure 6e,f) revealed that, in wall HI1, a moisture leakage of 6.9 g took 19 h and a moisture leakage of 20.7 g took 45 h, on average, to return the RH level near to the values before the leakages. For the other walls the return times were similar, although the changes in RH were substantially lower.

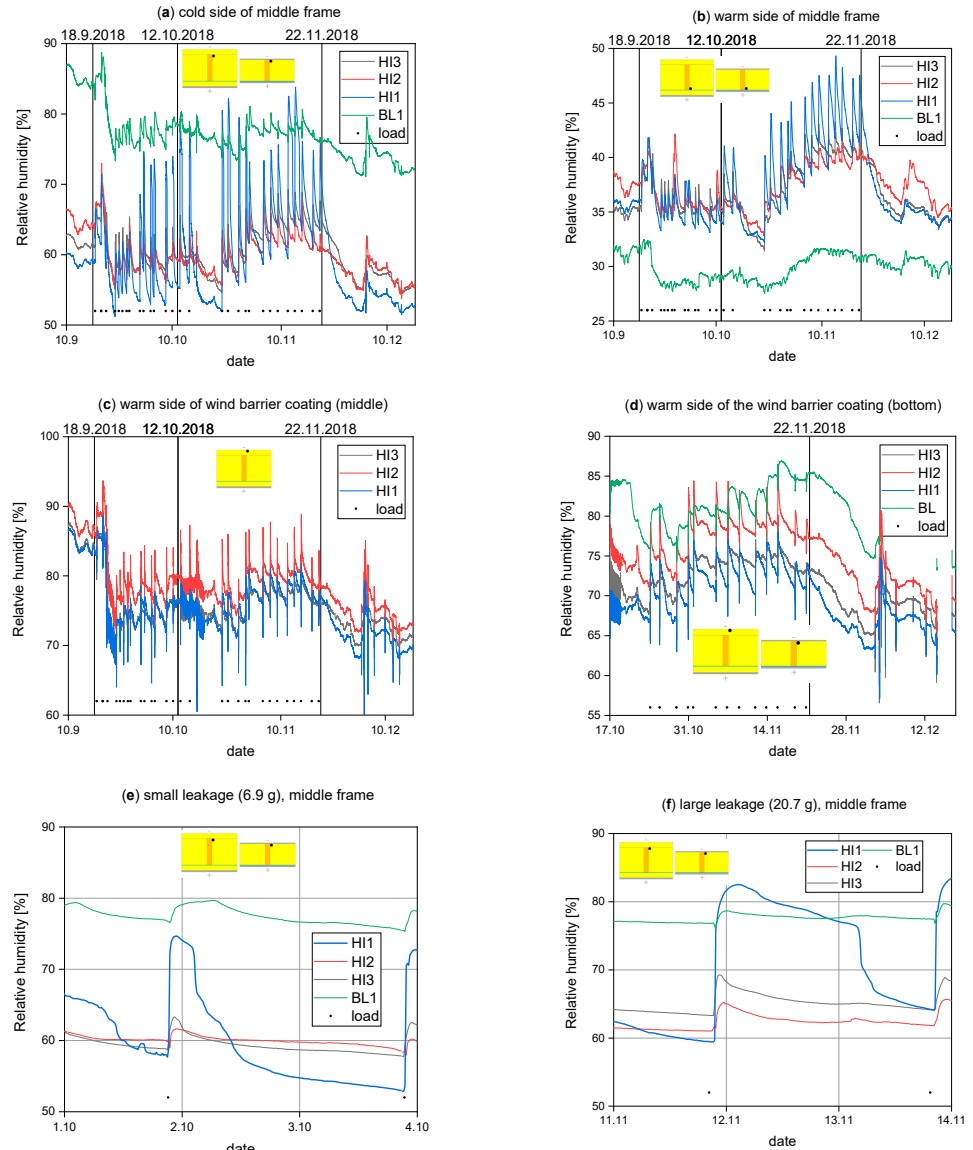

**Figure 6.** (**a**–**c**) relative humidity from the middle height of the wood frame during the leakage test; (**d**) additional measurement points were located 5 cm upwards from the bottom of the insulation; (**e**,**f**) an example of relative humidity change during small and large leakage events. The RH results are shown until 18 December 2018 because significant RH level changes were not observed afterwards.

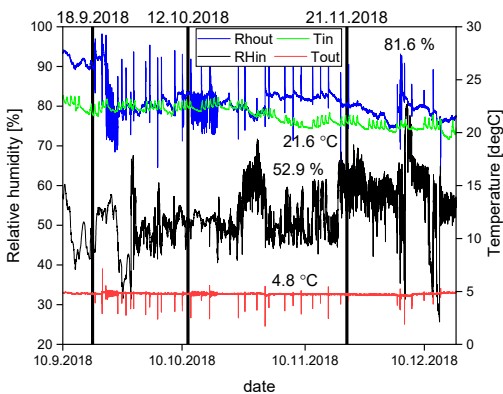

**Figure 7.** Climate conditions (T/RH) during the leakage test.

The warm side of the wood frame in the middle height (Figure 6b) had a relative humidity between 27–32% for the BL wall and 32–49% for the HI walls. Water leakage generated peaks in HI walls, but not for the baseline wall. This implies that the gypsum board in the BL wall moderated the RH changes in the warm side of the frame. The RH peaks were possibly evened by the PA-foil in wall HI2 and the gypsum board in wall HI3. In phase two, the average relative humidity rose in the BL wall from 29–31% and in the HI walls from 35–41%. The drying time, based on the RH levels, was similar to the cold side of the wood frame, which is consistent with the results from the built-in moisture test.

The outmost measurement points in the HI walls behind the wind barrier coating (Figure 6c) had relative humidities between 70% and 85%. The differences between the different HI walls were small. In addition, the RH level change between leakage phases was relatively low. However, an elevation of RH from 76–80% was present with the higher leakage rate. It is noteworthy that the RH level in the HI walls began to decline already before the last two leakage events, which is explained by the high drying ability of the structures originating from the low Sd-values presented in Section 3.1.

The additional T/RH measurement points in the bottom part of the frame (Figure 6d), added during leakage phase two, showed that the RH level was higher for the baseline wall compared to the middle height measurement. This might result from the sealing tapes of the test walls mentioned in Section 3.2. Walls HI1 and HI3 showed 4% lower RH results for the bottom location compared to middle height. A possible reason for this is the higher temperatures at the test wall edges, due to the plywood frame. The cold bridge of the plywood frames was discontinuous to the outdoor because wb-mineral wool was assembled over the plywood frame. The cold bridge was connected to the indoor because the soft 50 mm wool was assembled inside the plywood frame. What stands out was the increasing level of RH in the BL wall at the end of leakage phase two, which was not visible for HI1–HI3 walls. Thereby, moisture risks might appear before in the BL wall than in the HI walls, under high rain infiltration levels. The bottom part of the walls had similar drying curves compared to the middle height of the walls.

The results from the moisture content of the wood frames during the rain infiltration test are presented in Figure 8. The wood moisture results for the HI3 wall were omitted from the analysis, since these measurements suffered from significant electrical interference during the leakage test. This effect distorted the MC results for wall HI3 to be unusable. The interference effect was apparently lower in the other walls. The interference was caused by an electrical current near the datalogger of the MC electrodes. Therefore, only the electrodes in one logger signaled this effect. According to the manufacturer of the MC measurement equipment, the effect does not change the measurement equipment's reading accuracy after the interference has disappeared.

The wood moisture content in the middle height of the frame in the cold side (Figure 8a) showed minor changes connected to the water leakage test. In walls HI1 and HI2 moisture content rose 0.2–0.3 wt-% during the second leakage phase. The drying of built-in moisture was still present in the BL wall at the cold side of the frame because the MC curve declined throughout the test. The drying accelerated at the start of the leakage test because the level change in the outdoor RH (Figure 7). This effect was not present for the HI walls, possibly due to the R-value of the wb-mineral wool. The wind barrier mineral wool declined the RH level in walls HI1 and HI2. Therefore, the outdoor RH level change from 90–80% did not affect lower RH levels in the cold side of the wood frame.

At the warm side of the middle frame (Figure 8b) the moisture content increased less than in the cold side, approximately 0.05–0.15 wt-% in the HI walls and 0.175 wt-% in the BL wall. The MC in the BL wall was stable, which implies that the R-value level of the wb-layer and the decline of the outdoor RH do not affect the MC level of the wood frame in the warm side. The MC changes in the walls (Figure 8a,b) imply that the leakage moisture was directed into the middle height of the frames to a small extent. This is consistent with the assumption that most of the leakage moisture will drain to the bottom of the wall.

In the bottom part of the walls, the warm side of the frame (Figure 8d) did not react to the leakages at phase one while the cold side (Figure 8c) had a slight elevation (0.25–0.5 wt-%) in the moisture

contents of the frames. These MC results imply that the recurring 6.9 g water leakages do not rise severe moisture risks in the structures. In addition, the effect of recurring small moisture leakages is directed to the cold side of the wood frame.

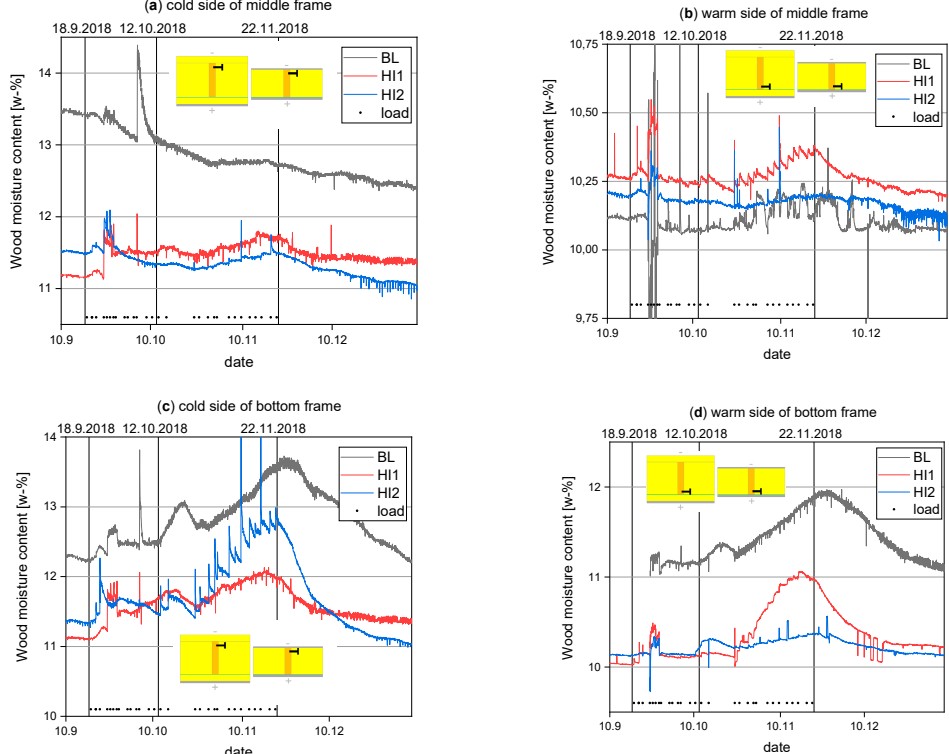

**Figure 8.** (**a**–**d**) Measured moisture content from the wood frame during the leakage test. Measurement locations are described in the subfigure headings.

In phase two there was two leakage events during the first four days, after which there was nine days break in leakages. This resulted in an increase in the moisture contents in the beginning, after which the fall in the moisture contents lasted until the leakages were started again (Figure 8c,d). The total MC change in the frame bottom for all walls was at the level of 0.8–1.4 wt-%. However, the warm side of wall HI2 experienced minor changes in moisture content, compared to the other results. The MC in the cold side of wall HI2 rose significantly compared to other measurement points in HI1–2 and the BL walls. Comparing MC changes in wall HI2 to the MC results from wall HI1, it is possible that the moisture transfer in wall HI1 was directed evenly to the bottom of the wood frame, while in wall HI2 moisture transfer was mainly directed to the cold side of the bottom frame. The PA-vapor barrier might somewhat reduce the MC level in the warm side of wall HI2. The drying rate was highest for wall HI2 in the cold side of the frame. Furthermore, the high wetting and drying rates in wall HI2 might result from an incomplete surface contact between the plywood frame of the test wall and the wood frame. This would explain the low MC changes in the warm side of the HI2 wall.

*3.5. Results from the Drying Simulations*

3.5.1. Model Validation

The model validation was made based on the relative humidity and moisture content in the measurement positions presented in Section 2.2. The validation results are presented in Figure 9. The relative humidity in the baseline wall differentiated, at most, 10% in the drying stage (Figure 9a,b). For wall HI1, the relative humidity differentiated, at most, 20% correspondingly. The differences in RH became small after 17–34 days. However, for the HI1 wall a difference of 8% stayed between simulated

and measured RH at the cold side of the frame (Figure 9a). This could be connected to the differences in heat transfer between the simulation and the built-in moisture test. However, the actual simulations were made with similar heat transfer properties, which leads to comparable results between simulation cases. The higher simulated RH, compared to the measured value, results in secure results when evaluating moisture safety of structures. In the drying phase, the simulated moisture content varied according to depth of the wood, after which the moisture content became steady and independent on the wood depth (Figure 9c–f). This shows how the initially unevenly moist wood frame equalizes the MC during the drying period. At the cold side of the frame the simulated MC was higher than the measured MC (Figure 9c,e). At the warm side of the frame simulations were more accurate and the MC was mostly below the measured MC (Figure 9d,f).

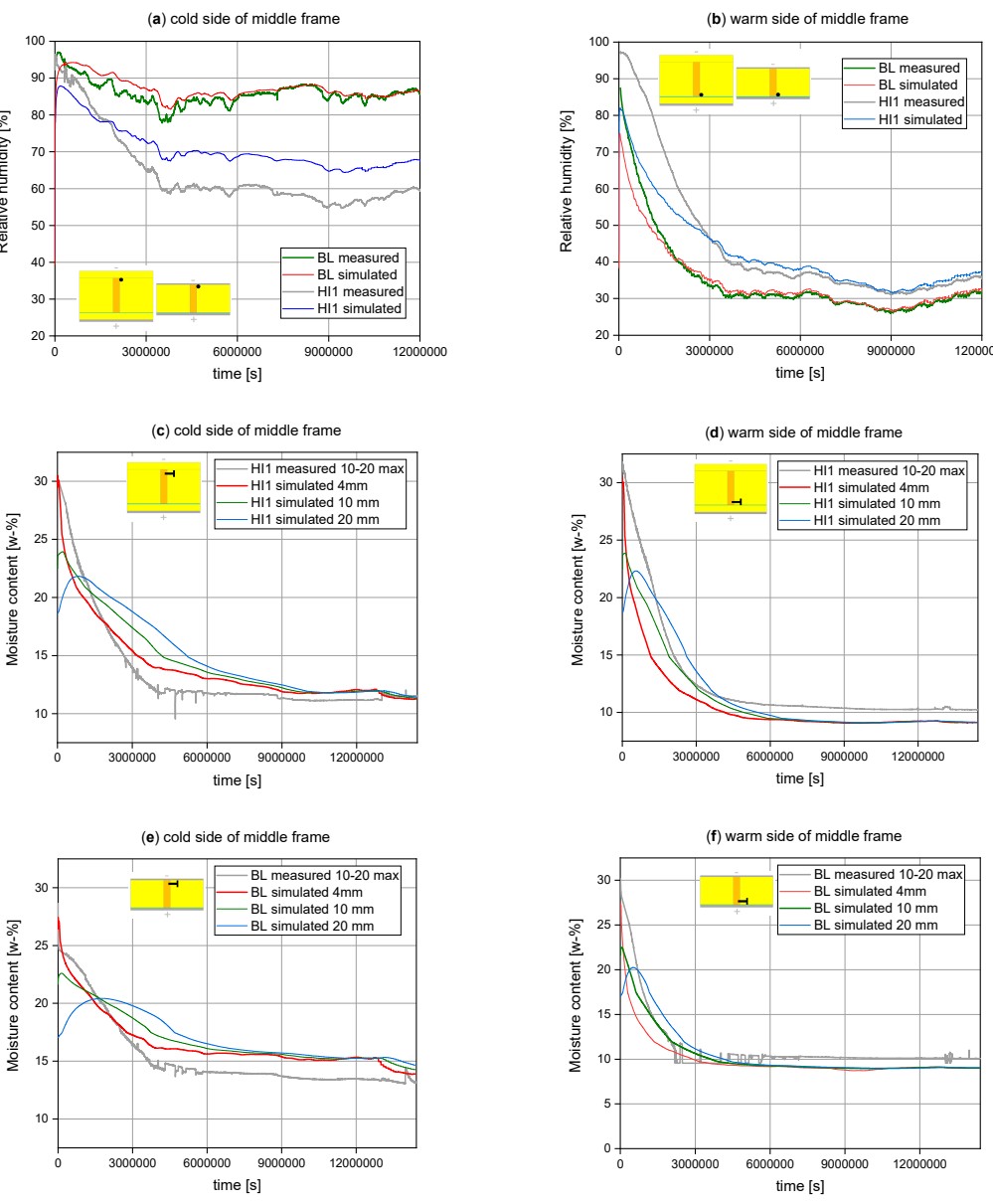

**Figure 9.** (**a**,**b**) the relative humidity and (**c**–**f**) the wood moisture content results from the simulation model of the BL and HI1 walls, compared to the measurement values. The moisture contents are presented in three different depths in the wood frame, whereas the measurement represents the maximum value in the depths between 10–20 mm. Measurement locations are described in the subfigure headings.

There are many factors that affect the similarity of simulated and measured RH and wood MC. Relative humidity is closely related to temperature; therefore, the aforementioned differences in heat transfer in simulated and measured structures is one factor. Especially for wood MC, the initial moisture level plays a major role. In the present experimental test, the initial moisture in the frame was strongly uneven. Most of the moisture was in the outer part of the frame. For calculation purposes, this was simulated with a limited amount of wood layers. The second factor affecting the MC results is the wood sorption curve, which can vary a lot, even for samples from same wood species. The wood planks used in the experimental walls had densities between 367–437 kg/m$^3$ while the average density of the sorption test samples was 514 kg/m$^3$. Another factor is the moisture measurement principle. The measurement depth can vary at least $\pm 1$ mm. Also, the measurement indicates the maximum moisture content between the electrodes. Considering all these error sources, the observed deviances were understandable, and the validation results were considered acceptable.

### 3.5.2. Results from the Drying Simulations

The simulation results from the drying phases are presented in Figures 10–12. The simulation time was 5500 h (229 days). The average moisture content of the wood frame was monitored during the drying phase. In HI5 and HI6 structures, this value represents the outmost 200 mm part of the 50 × 350 mm$^2$ frame and, in HI7, this value represents the moisture content of the outer frame (Appendix A).

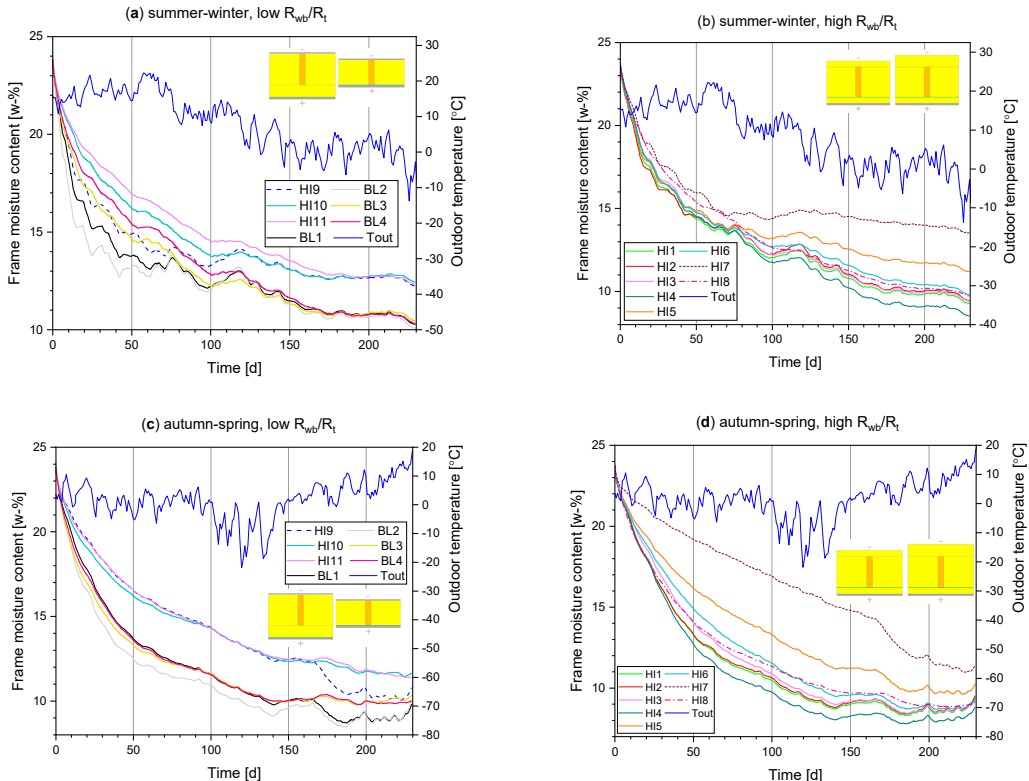

**Figure 10.** (**a**–**d**) The moisture content results from the drying simulations. The simulation started during summer (**a**,**b**) and during autumn (**c**,**d**). The simulation results are divided in two subfigures based on the thermal insulation level of the weather barrier layer (low thermal insulation level (**a**,**c**), high thermal insulation level (**b**,**d**)).

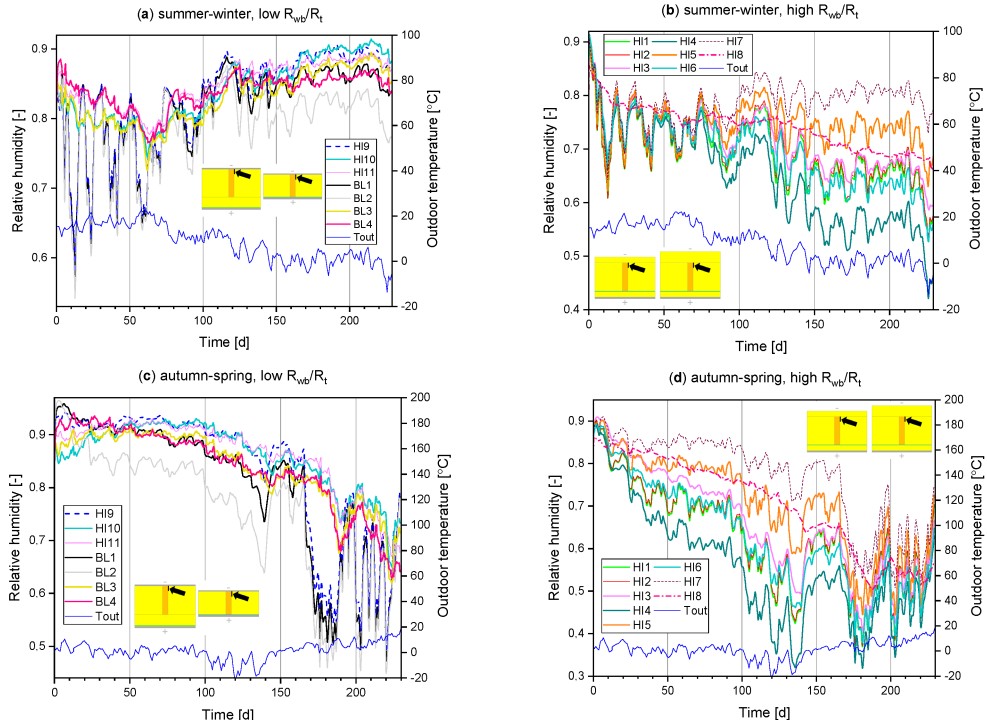

**Figure 11.** (**a**–**d**) The relative humidity results from the drying simulations. The simulation started during summer (**a**,**b**) and during autumn (**c**,**d**). The simulation results are divided in two subfigures based on the thermal insulation level of the weather barrier layer (low thermal insulation level (**a**,**c**), high thermal insulation level (**b**,**d**)).

Differences in the drying rates were observed during the summer-winter simulation (Figure 10a,b). During the first 100 days, wall HI11 had the lowest drying rate. At the end of simulation, the moisture loss was lowest in the HI7 wall with 13.5 wt-% moisture content in the frame. Walls HI9–11 had the second lowest drying rates, ending with 12.3 wt-% MC. Wall HI5 had the fifth lowest drying rate, with a final MC of 11.2 wt-%. The final MC for the other HI and BL walls varied between 8.5–10.5 wt-%. The drying rate in wall HI7 was deteriorated during the cold season when the drying almost ceased. This could be connected to the hygrothermal behavior of this extremely highly insulated wall (650 mm); the temperature in the warm side of the wb-mineral wool is near the outdoor air temperature despite the relatively high thermal resistance outside the frame. In walls HI9–HI11, with 300 mm insulation thickness, the drying rate was higher than in wall HI7. The same applied for wall HI5, with 450 mm insulation thickness and the same wb-material and thickness as in wall HI7. These results suggest that, with high total insulation levels reaching 650 mm, the thermal resistance of 50 mm mineral wool in the wb-layer does not guarantee the highest drying rates. Furthermore, the thermal resistance of the wb-layer should probably be some minimum proportion of overall R-value of the wall structure. This is evident from the low final MC results in walls HI1–4. Wall HI4 had the highest drying rate, with a final moisture content of 8.5 wt-% and walls HI1–3 finished second with MC between 9.3–9.4 wt-%. The $R_{wb}/R_t$ -ratios (Table A2) were 20% for walls HI1–3 and 33% for wall HI4. For wall HI7, the $R_{wb}/R_t$ -ratio was 9% and for wall HI5 it was 14%. The drying of walls HI1–3 was almost identical. This suggests that the PA-foil as vapor barrier or gypsum behind wb-mineral wool does not affect drying rates. The wall HI6 was the second wall with 100 mm wb-insulation, in addition to wall HI4. This did not ensure as high a drying rate as in wall HI4, which could result from the higher overall insulation thickness (450 mm) in wall HI6. However, the drying rate in wall HI6 was higher compared to wall HI5, both with same total insulation thickness. This further supports the positive effect of higher $R_{wb}/R_t$ -ratio on the drying rates.

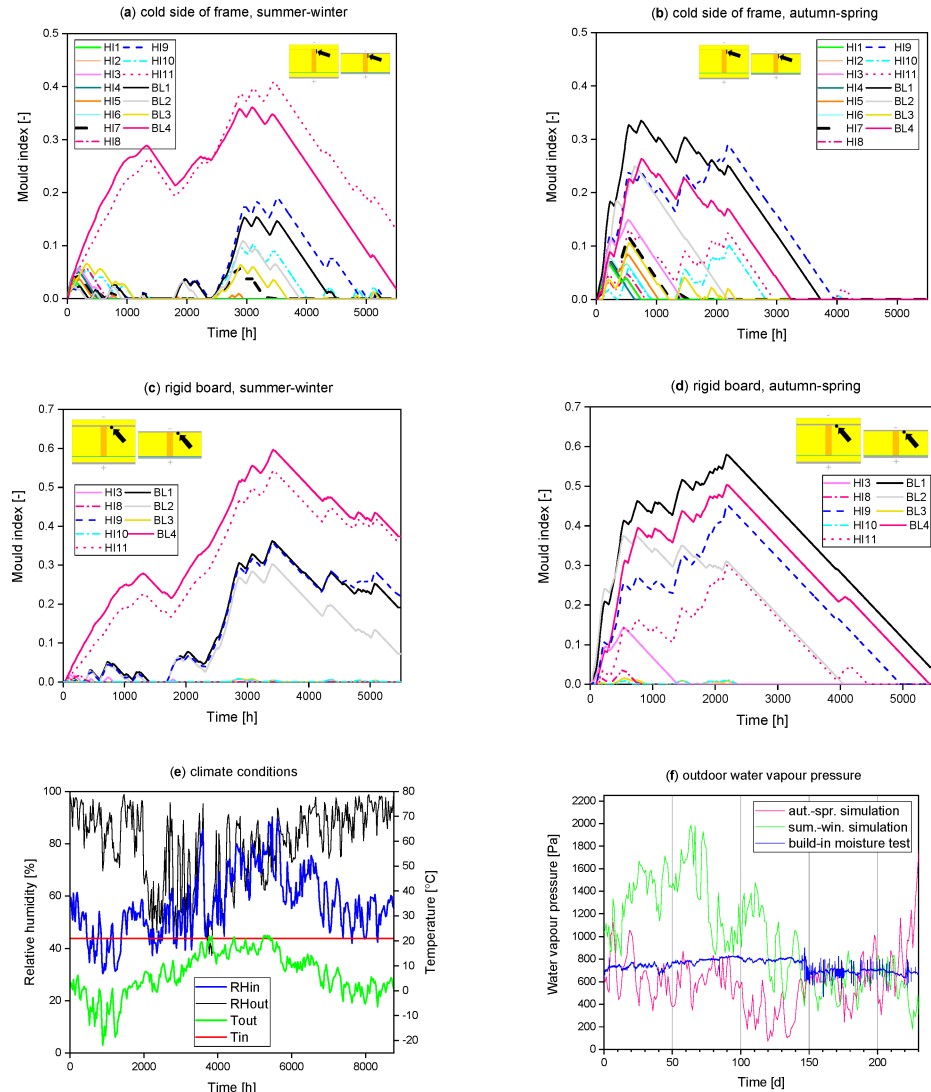

**Figure 12.** (**a**–**d**) Mould index results; (**e**) weather conditions; (**f**) outdoor water vapor pressure during the drying simulations. Mold index was determined on the cold side of the frame (**a**,**b**) and on the warm side of the rigid board (**c**,**d**).

During the first 70–80 days of the summer-winter simulation, walls BL1–2 had the highest drying rates (Figure 10a). After 50 days of simulation the MC in walls BL1–2 was 13.2–13.8 wt-% whereas in other walls the MC varied between 14.5–17.0 wt-%. The final MC in walls BL1–2 was 10.0–10.3 wt-%, which was at the same level as in the other BL walls. Apart from walls HI5 and HI7, the final MC for HI walls with high $R_{wb}/R_t$ -ratios was slightly lower (9.3–9.7 wt-%) than in the BL walls. The final MC in walls HI9–11, with low $R_{wb}/R_t$ -ratios, was higher than in the BL walls. This suggests that the U-value of the wall structure (Table A2: BL walls 0.21–0.36 W/m²K, HI walls 0.13 W/m²K) has an impact on the drying ability when the $R_{wb}/R_t$ -ratio is low.

In the summer-winter simulation, relative humidity declined under 85% in every wall during the first 30 days of simulation (Figure 11a,b). The RH levels were highest at the beginning of the simulation in walls BL4 and HI11, probably due to the Sd-value in the plywood (0.2–2.1 m). During winter, the relative humidity rose in walls with low thermal insulation outside the frame to a level of 85–90%. In wall BL2 relative humidity was at level 80–85 due to the highest U-value. The RH level in the HI walls with high $R_{wb}/R_t$ -ratios continued to decline, even during winter. Walls HI4 and HI6 had the lowest RH during winter, resulting from the 100 mm wb-mineral wool. Walls HI11 and BL4 showed the highest mould index levels on the frame surface, reaching a level of 0.3–0.4 and on the rigid board

0.5–0.6 (Figure 12a,c). For other walls, notable MI levels were observed only during winter. Particularly, walls BL1 and HI9 had MI values around 0.15 on the wood frame and 0.35 over the gypsum board. BL2 had notable MI value around 0.3 over the gypsum board. At the end of the simulation, the MI values were elevated on the wood frame of wall HI11 and on the rigid boards in walls HI9, HI11, BL1–2, and BL4. Despite the RH near 90% at the end of simulation, the MI level was low at wall BL3, probably due to the mould resistance of the FCB.

In the autumn-spring simulation, wall BL2 had the highest drying rate during the first 50 days, after which wall HI4 dried the fastest, reaching a level of 8.8 wt-% at the end of the simulation (Figure 10c,d). Walls HI1–3 had final MC of 9.3–9.5 wt-% and the drying was as fast as in the summer-winter simulation. Walls HI6 and HI8 had also low final MC of 9.1–9.3 wt-%, which were slightly lower MC values compared to the summer-winter simulation. Highly insulated walls HI5, HI7, and HI9–11 had the lowest drying rates. The probable reasons for the lower drying rates in these walls are the increased insulation level from 300 mm to 450–650 mm (HI5, HI7) and the low $R_{wb}/R_t$ -ratios (HI9–11). At the end of the simulation drying increased in HI9 and HI7, walls with rising outdoor temperature resulting in final moisture contents of 11 wt-%. Walls HI10–11 did not reach the same way, which could result from the more vapor tight FCB and plywood boards, compared to the gypsum board. This is supported by the low reaction in MC with rising outdoor temperature in wall HI8 and the higher reactions in walls which had gypsum as wb-layer. These results imply that vapor open wb-layer allows the structure to dry faster when the outdoor temperature rises. Further, the low drying of extremely highly insulated wall HI7 in autumn (Figure 10d) and in winter (Figure 10b) indicates that the drying rates in extremely highly insulated structures are impaired by cold weather. One reason for this is that the temperature difference across the wb-layer during cold weather is smaller, with higher U-values of the structure.

The lowest RH levels during the autumn-spring simulation were observed for walls HI1–4 and HI6 (Figure 11c,d). The remarkably low RH in wall HI4 originated from the highest $R_{wb}/R_t$ -ratio of all the walls. Walls HI5 and HI7 had higher RH values due to the lower $R_{wb}/R_t$ -ratio than in HI1–4 and HI6. The higher RH level in wall HI8, compared to similar HI walls without the rigid board, was probably connected to the additional Sd-value from the FCB (0.15–0.4 m). However, the mould index was not increased probably due to the resistance against mould growth in FCB. For walls BL1, BL3, BL4, and HI9–11, the relative humidity stayed at a 90% level for approximately 100 days. For wall BL2, the corresponding RH value was 85%, probably due to the higher U-value. These RH levels were noticeable in the mould index on the wood frame, reaching 0.2–0.3 values in walls BL1, BL2, BL4, and HI9. The frame mould indexes declined to zero for all walls after 4000 h of simulation. The mould indexes over the rigid boards in these walls were higher, reaching values of 0.3–0.58 with a similar but slower decrease towards the end of the simulation. The observation point on the rigid board has a lower temperature than the surface of the wood frame and this point is in the probable path of the highest water vapor diffusion rates from the wood frame during the drying process. These factors might make the rigid board surface more probable to mould growth. The relative humidity declined by several tens of percent in most of the walls during the spring when the outdoor temperature rose to 0–10 °C. Walls BL3, BL4, HI10, and HI11 did not react similarly. A lower decrease in RH was observed after approximately 20 days from this. A probable reason for this is the use of more vapor tight FCB and plywood as the wb-layer. However, the lower decrease in RH did not affect the MI results as the RH level was already at 80% in these walls.

## 4. Discussion

In this paper, the drying process of a wood frame walls with high insulation levels has been investigated. The main objective was to examine whether the R-value of the wall influences its drying ability. Experimental tests were set up with a high build-in moisture levels, followed by simulation analysis, which was used to evaluate the drying process in the Finnish climate.

The wetting behavior of the wood disks in the drying cakes test showed that wood takes moisture rapidly from the surrounding humid air. This must be considered in the construction site. Wood frames and other wood based structural parts should be weather protected. In details where the drying ability is weaker, the level of built-in moisture must be kept low.

The drying cakes test indicated that the lower R-value of the baseline walls did not ensure faster drying rates, compared to the highly insulated walls. In fact, the highly insulated walls had faster drying rates throughout the test even though the Sd-value of the outer part of the BL walls was roughly half of the Sd-value of the HI walls. The largest drying rate difference during phase one was probably connected to temperature differences between the HI and BL walls. The thermal resistance of the wind barrier layer in the BL walls was around 1% and 17–22% in the HI walls from the total thermal resistance of the structures. Therefore, the water vapor pressure difference between the outer surface of wood disks and the outdoor air was higher for HI walls. Later, the drying rates became more alike as the drying was more and more influenced by the moisture transfer inside the wood disks. The results indicated that drying ability in in baseline walls with low thermal resistance is slightly increased with larger U-values. It must be considered that the test assembly did not allow lateral moisture transfer. Moisture transfer from a moist wood frame wall is presumably at least two-dimensional in nature. Therefore, the drying results apply to a structural detail without lateral moisture transfer.

The wetting phase of the wood frames indicated a strong radial difference in the wood moisture levels. The core of the wood stays drier while the outer surface is highly moistened. The first few millimeters from the surface has the highest moisture content, reaching up to 34 wt-%. These wetting properties probably accelerate the drying process of the wood frame wall, due to the shorter distance of the moisture to reach the wood surface.

All the walls showed a high drying rate during the built-in moisture test. At the cold side of the frame, the relative humidity declined slower, possibly due to the lower vapor pressure difference to the outdoor air. The higher RH for the BL wall resulted from the low R-value of the gypsum board. Similarly, the lower RH at the warm side of the BL wall compared to the HI walls resulted from the low R-value of the structure inside the wood frame. The results suggest that using gypsum board behind wind barrier mineral wool has a small effect on the drying rates. However, the minor effect does not rise moisture risks in highly insulated wall structures.

At the bottom part of the wall, the low drying rate at the cold side of the BL wall was probably caused by the sealing tape used in the test structure and the high initial moisture content in the cold side of the wood frame in the BL wall. The sealing tape was assembled 20–25 mm over the lower edge of the wind barrier gypsum board. In HI walls this taping did not affect the drying since the 50 mm wind barrier insulation board allowed moisture to dry in the vertical direction. At the warm side of the frame this masked edge type of sealing did not affect drying rates, since moisture transfer spread to a larger surface area. The MC results showed high initial moisture content (over 35 wt-%) for the cold side of the bottom part of the BL wall, which indicated that the initial moisture content in the frame was not even. This reduced the drying rate in the BL wall even more. Furthermore, the drying rates between HI walls showed that the PA-vapor barrier did not enhance the drying rate, which is consistent with an earlier study [27]. This was due to the high drying rate towards outdoor and elevated indoor moisture levels. The highest RH behind the wind barrier coating in wall HI2 was probably caused by the lower Sd-value of the vapor barrier, resulting in a higher moisture diffusion rate from the inside air.

The results from the rainwater infiltration test showed that the small leakage moisture inserted into the insulation cavity of a wood frame wall did not have an adverse effect to the hygrothermal behavior of the structures. Relative humidity varied between 55–85% in the outer part of the structures. This indicates that the fault tolerance of vapor open (Sd 0.07–0.17 m) baseline and highly insulated structures against leakage moisture was at a high level. With a larger leakage moisture, an increasing trend in the RH and MC values was evident for most of the measurement points in the walls. Further, the level of the RH and MC changes indicated that most of the leakage moisture still dried out of

the test structures. However, the experiment results imply that recurring larger leakage events lead into ascending moisture contents in the structure and eventually into moisture damages. The thermal insulation level of the structures did not significantly affect these tendencies in the behavior of the structures. At the bottom part of the wall, the risk for moisture damages can be slightly higher for baseline walls compared to highly insulated walls. The results imply that PA-vapor barrier and gypsum board behind wb-mineral wool or as the weather barrier layer can, in the short term, equalize relative humidity changes in the insulation cavity, connected to rainwater infiltration.

The validation of the simulation model demonstrated that it is possible to evaluate the drying process of a moist wood frame with acceptable accuracy by adoption of correct water vapor diffusion coefficients. The diffusion coefficient of wood must include the strong dependency on the relative humidity. The simulation results indicated that the drying ability of the wood frame wall is not remarkably different between baseline and highly insulated walls. If the drying will take place during summer, a BL wall with 100–175 mm mineral wool insulation can dry slightly faster compared to a HI wall with 300 mm insulation, during the first 1–2 months. If the drying starts in autumn, the drying rates are almost identical between walls with different insulation levels. This behavior can be associated with the low water vapor pressure in the outdoor air. With a high drying potential, the drying rate of the wood frame is determined with the moisture transfer inside the wood frame. The highest drying rates can be obtained with highly insulated structures, using moderate insulation levels (300–350 mm), and by using 50–100 mm mineral wool outside the wood frame.

The risk for mould growth during the drying phase was low for all walls. The maximum value of the mould index, approximately 0.6, was found on the rigid boards of walls BL1 and BL4, which is below the limit of 1.0, standing for growth detectable by microscope. The most common highly insulated wall type, HI1, had very low mould index (0.065) and, in this sense, performed better than the BL1 wall. The results implied further, that a plywood weather barrier might cause moisture problems in structures with high built-in moisture levels, regardless of the U-values of the structures. A gypsum board as weather barrier can lead to slight mould growth, especially if the drying phase starts in autumn. If using a thermal insulation as wind barrier, the risk is diminished. The third investigated rigid board, a fiber cement board, did not imply mould risks due to its the resistance against mould growth. The lowest RH, in wall HI4, implied that the thermal resistance on the cold side of the frame enhances the hygrothermal behavior of the wall by ensuring lower RH in the outer part of the structure. The results did not indicate severe moisture risks, even with insulation thickness of 650 mm. However, the drying rates in extremely highly insulated structures are not as high as in highly insulated structures (insulation 300–350 mm). Cold weather, especially, can significantly lower the drying rate in extremely highly insulated structures.

## 5. Conclusions

This paper focused on the moisture safety related to energy efficient wood frame wall structures. The objective of this study was to evaluate whether the R-value of the wall structure influences the drying-out ability of the structure. The need for the research was based on the controversial conclusions in the earlier studies, concerning the possible moisture risks with highly insulated structures. The research evaluated the drying-out ability and the hygrothermal behavior of the wall structure based on two significant moisture sources, built-in and leakage moisture.

The results suggest that highly insulated wall structures promote healthy buildings, as their drying ability under severe moisture loads is at a high level. High mould indexes during the drying phase in highly insulated walls are unlikely, even with a considerable amount of built-in moisture or rain leakages. The moisture problems reported earlier concerning the hygrothermal behavior of highly insulated walls under water leakages are probably connected to the use of vapor tight materials, such as medium density fiberboard (MDF), outside the wood frame. A more vapor open wind barrier mineral wool outside the wood frame was used in this paper. The use of vapor tight rigid boards outside the wood frame must be avoided. The use of plywood as a weather barrier in structures

with build-in moisture might result in slight mould growth. With rigid boards, the use of a thermal insulation board as the outmost layer will reduce the moisture risks. A vapor open thermal insulation board with a thickness of 50–100 mm, installed on the cold side of the wood frame, contributes to an increased drying ability of the wall and decreased probability for microbial growth in the structure, which is consistent with earlier studies.

The study results support the following topics for future research:

- Study the hygrothermal behavior of highly insulated wood frame structures with significant moisture loads, further considering moisture convection and water vapor diffusion from elevated indoor humidity levels;
- Detailed analysis on the moisture transfer in highly insulated structures after water leakages;
- Study the influence of more hygroscopic thermal insulation, for example wood fiber insulation, to the drying ability of highly insulated structures under high moisture loads.

**Author Contributions:** Conceptualization K.V. and X.L.; methodology, validation, formal analysis, visualization, and writing—original draft preparation K.V.; project administration, writing review, and editing X.L.

**Funding:** This research was funded by the Foundation for Aalto University Science and Technology. In addition, the research was funded with the PhD-scholarship from SNIL-scholarship Fund.

**Acknowledgments:** The authors would like to thank the Foundation for Aalto University Science and Technology and SNIL ry for the funding of this study.

**Conflicts of Interest:** The authors declare no conflict of interest. The funders had no role in the design of the study; in the collection, analyses, or interpretation of data; in the writing of the manuscript; or in the decision to publish the results.

## Appendix A

The specifications for the structures used in the simulations described in Section 2.4 are presented in Table A1. The thermal properties of the structures are presented in Table A2. The geometries of the 2D simulation models are presented in Figure A1.

**Table A1.** Material layers in the simulation models. Material layers are listed starting from inside to outside of structures.

| | HI1, HI2 | HI3 | HI4 | HI5 | HI6 |
|---|---|---|---|---|---|
| Frame | 50 × 200 | 50 × 200 | 50 × 200 | 50 × 350 | 50 × 350 |
| Material layers | Gypsum 13<br>GW 50 [1]<br>PE-foil/PA-foil (HI2)<br>GW200 + frame<br>GW50 (wb) | Gypsum 13<br>GW 50 [1]<br>PE-foil<br>GW200 + frame<br>Gypsum 9 (wb)<br>GW50 (wb) | Gypsum 13<br>GW 50 [1]<br>PE-foil<br>GW200 + frame<br>GW100 (wb) | Gypsum 13<br>GW 50 [1]<br>PE-foil<br>GW350 + frame<br>GW50 (wb) | Gypsum 13<br>PE-foil<br>GW350 + frame<br>GW100 (wb) |
| | **HI7** | **HI8** | **HI9, HI10, HI11** | | |
| Frame | 2 × (50 × 200) | 50 × 200 | 50 × 200 | | |
| Material layers | Gypsum 13<br>PE-foil<br>GW200 + frame<br>GW200<br>GW200 + frame<br>GW50 (wb) | Gypsum 13<br>GW 50 [1]<br>PE-foil<br>GW200 + frame<br>Fiber Cem. 9<br>GW50 (wb) | Gypsum 13<br>PE-foil<br>GW 100 [1]<br>GW200 + frame<br>Gypsum 9 (wb)—HI9,<br>or Fiber Cem. 9—HI10,<br>or Spruce Plywood 9—HI11 | | |
| | **BL1** | **BL2** | **BL3** | **BL4** | |
| Frame | 50 × 175 | 50 × 100 | 50 × 175 | 50 × 175 | |
| Material layers | Gypsum 13<br>PE-foil<br>GW175 + frame<br>Gypsum 9 (wb) | Gypsum 13<br>PE-foil<br>GW100 + frame<br>Gypsum 9 (wb) | Gypsum 13<br>PE-foil<br>GW175 + frame<br>Fiber Cem. 9 | Gypsum 13<br>PE-foil<br>GW175 + frame<br>Spruce Plywood 9 | |

[1] Thermal conductivity increased from 0.033 W/mK to 0.038 W/mK due to wood frame.

**Table A2.** Thermal properties of the structures in the simulations.

|  | **HI1–3** | **HI4** | **HI5** | **HI6** | **HI7** | **HI8** | **HI9+11** | **HI10** | **HI11** |
|---|---|---|---|---|---|---|---|---|---|
| U-value [W/m$^2$K] | 0.12 | 0.10 | 0.08 | 0.08 | 0.06 | 0.13 | 0.13 | 0.13 | 0.13 |
| $R_{wb}/R_t$ [%] | 20 | 33 | 14 | 27 | 9 | 20 | 0.8–1 | 0.3 | 1 |

|  | **BL1** | **BL2** | **BL3** | **BL4** |
|---|---|---|---|---|
| U-value [W/m$^2$K] | 0.22 | 0.36 | 0.22 | 0.21 |
| $R_{wb}/R_t$ [%] | 0.9 | 1.6 | 0.6 | 1.6 |

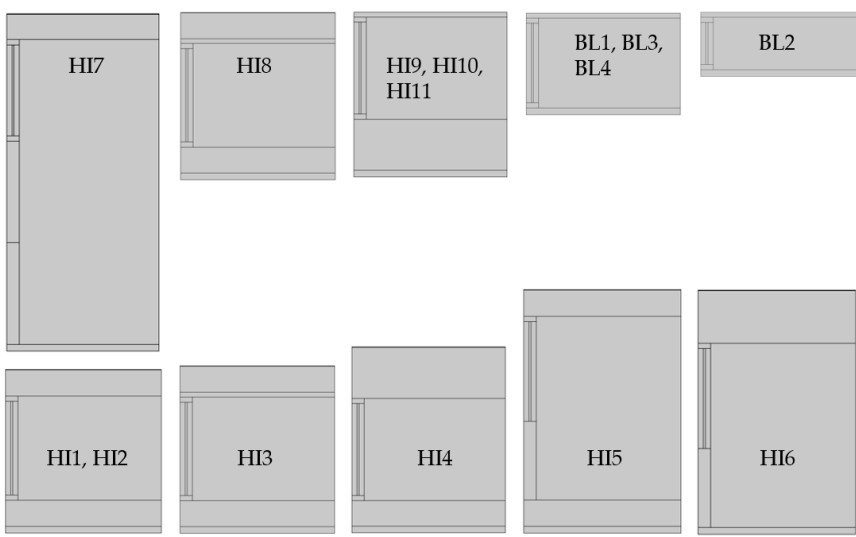

**Figure A1.** The simulation geometries of the wall structures.

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
