# Peer review of "An Experimental Study on the Drying-Out Ability of Highly Insulated Wall Structures with Built-In Moisture and Rain Leakage"

_applsci, doi:10.3390/app9061222_

Round 1
Reviewer 1 Report
The paper addresses the issue of drying-out ability of highly insulated wall structures with built-in moisture and rain leakage. The manuscript is worthy of publication. The results of the experiments were presented clearly, however, the reasoning behind different behaviours of materials in most sections of the result is missing. Authors must provide potential readers reasoning behind different behaviours of specimens in each section of "Result".
Author Response
Response to Reviewer 1 Comments
Point 1: The paper addresses the issue of drying-out ability of highly insulated wall structures with built-in moisture and rain leakage. The manuscript is worthy of publication. The results of the experiments were presented clearly, however, the reasoning behind different behaviours of materials in most sections of the result is missing. Authors must provide potential readers reasoning behind different behaviours of specimens in each section of "Result".
Response 1: The Section 3 (Results) has been revised with comparative analysis of the behaviour of the different materials and wall structures as requested.
Please see the manuscript for these and other changes based on comments from reviewer 2 and my own observations during the revision. I have used the Track Changes function in the manuscript.
Reviewer 2 Report
The submitted paper concerns the drying process of wood frame walls investigated in different seasonal environments. Several configurations with high insulation have been taken into account and compared to the control wall. A simulation model was also used and validated.
The study is properly organized and presented. Just a few aspects need to be clarified and revised.
- Table 1: Why different moisture amounts have been added during humidification? Might these differences affect the results?
- Line 242: Reference 26 is cited before than references 21-25.
- Line 252: Why winter has been considered as a warm season?
- Figures 3, 7, and 10: The graphs are not clear. It is very difficult to link each curve to the related parameters.
- Figures 3-6 and 8-11: In the caption, describe each graph. The use of subfigures (a), (b), etc. could help.
- Lines 340-360: The results should be discussed referring to the weathering conditions instead of dates.
- Lines 571-574: In my opinion, these statements should be reported in the discussion of the results; on the other hand, the microbial growth is not mentioned before. In addition, I suggest avoiding the use of references’ numbers in the conclusions section.
- Acronyms and symbols should be defined at their first appearance. Here a list of examples, indicative but not exhaustive: U-values (line 56), MC (line 138), Sd (Table 1), GW and wb (Table 2), and so on.
- Check the sections’ numbering.
Author Response
Response to Reviewer 2 Comments
The submitted paper concerns the drying process of wood frame walls investigated in different seasonal environments. Several configurations with high insulation have been taken into account and compared to the control wall. A simulation model was also used and validated.
The study is properly organized and presented. Just a few aspects need to be clarified and revised.
Point 1: Table 1: Why different moisture amounts have been added during humidification? Might these differences affect the results?
Response 1: The different additional moisture amount in the frames was not intended. The wood frames had to be kept in the weather room until the assembly of the test walls to prevent premature drying. It was not possible to precisely control the additional moisture amount in the frames. The reason for the different additional moisture amount in the frames is probably connected to the difference in the moisture intake properties of the frames. The frame density was 69 kg/m3 higher for BL frame compared to HI frames. This has probably an effect on the capillary moisture transfer properties of the frames. The difference in the initial built-in moisture level might have influenced the MC-results in the bottom part of the frame. Therefore, this has been commented in the text on lines 344-346, 706-708 and 712-714.
Point 2: Line 242: Reference 26 is cited before than references 21-25.
Response 2: The reference numbers have been corrected.
Point 3: Line 252: Why winter has been considered as a warm season?
Response 3: This was not intended. The sentence has been rewritten to express the weather conditions during the simulation more precisely.
Point 4: Figures 3, 7, and 10: The graphs are not clear. It is very difficult to link each curve to the related parameters.
Response 4: The figures 3, 7 and 10 (new figures 10 and 11) have been revised to present the results more clearly; Figure 3: The font sizes in legends were increased. Major ticks in the upper horizontal border were removed for the subfigure “(a) climate conditions”. The relative humidity value “88.6 %” was moved to the right in subfigure (a). The places for Y-axises were switched to match the rest of the article in subfigure (a). The unit for Y-axis Temperature was changed from “degC” to “ºC”. Added outdoor relative humidity graph to subfigure “(d) warm side of the wind barrier coating”. Maximum value for relative humidity and temperature axises were lowered in subfigure (d). Lineweights with value 2 were lowered to 1,5 to clarify the appearance of the graphs. However, baseline lineweights were kept at value 2. Line colors and linetypes were partially changed. Legend text “Outdoor” changed to “RHout” in subfigures (c) and (d).
Figure 7: The font size in legend was increased. The minor ticks in y-axis for relative humidity were removed.
Figure 10: The results are divided into two Figures 10 (MC) and 11 (RH). The results are presented separate subfigures for structures with high Rwb/Rt and for structures with low Rwb/Rt. The small structure figures were updated according to this division. In addition, all lineweights with value 2 were lowered to 1,5 to clarify the appearance of the graphs. The font sizes in legends were increased. The maximum value for relative humidity axis was lowered in Figure 11c,d.
Point 5: Figures 3-6 and 8-11: In the caption, describe each graph. The use of subfigures (a), (b), etc. could help.
Response 5: Subfigure labelling was added to figures 3-6 and 8-10 (figure 10 was split to figures 10 and 11) and 11 (new figure number 12). All captions were updated to describe the subfigures. The headings indicate the vertical location of results; thereby texts “BOTTOM” and “MIDDLE” were removed from figures 4, 6, 8 and 9.
Other revisions connected to the Figures:
-Added “and temperature” in the caption for figure 3.
-Text size was increased in legend of subfigure (b) in figure 5.
-Minor horizontal gridlines were removed from the subfigures (e) and (f) in figure 6. Text size in legend of subfigures (e) and (f) was increased. Locations of the legends in subfigures (e) and (f) were changed.
-Caption in figure 8 changed; “during the leakage test phases 1-3” changed to “during the leakage test”.
-Text size was increased, and empty spaces removed in legends of subfigures (a) and (b) in figure 9.
-Text size was increased in legend of all subfigures in figure 11 (new figure number 12). Major and minor ticks were removed from the upper horizontal borderline in subfigure (e).
Point 6: Lines 340-360: The results should be discussed referring to the weathering conditions instead of dates.
Response 6: The text has been revised; the dates have been replaced with the thermal condition during the test. Starting from line 383 (cakes) and 417 (rain infiltration) the dates have been taken out. The phase numbers in the drying cakes test have been added to subfigure (b) in Figure 5 to concentrate on the phases instead of dates.
Point 7: Lines 571-574: In my opinion, these statements should be reported in the discussion of the results; on the other hand, the microbial growth is not mentioned before. In addition, I suggest avoiding the use of references’ numbers in the conclusions section.
Response 7: The text starting from line 768 has been changed; the “microbial growth” has been removed to avoid confusions. Reference numbers have been removed from Section 5 (Conclusions).
Point 8: Acronyms and symbols should be defined at their first appearance. Here a list of examples, indicative but not exhaustive: U-values (line 56), MC (line 138), Sd (Table 1), GW and wb (Table 2), and so on.
Response 8: The definions of acronyms and symbols have been added to the first appearance in the text. The word order in the first definitions have been changed in corresponding locations; eg. at line 179 “the DRF (the driving rain factor) set to 0.225” was corrected to “the driving rain factor (DRF) set to 0.225”
Point 9: Check the sections’ numbering.
Response 9: The section numbering has been corrected; 3.4. Results from the rain infiltration test (old numbering 3.3).
Please see the manuscript for revisions from points 1-9 and other changes based on comments from reviewer 1 and my own observations during the revision. I have used the Track Changes function in the manuscript.
Round 2
Reviewer 1 Report
Comments were reasonably addressed.